# Pain from Internal Organs and Headache: The Challenge of Comorbidity

**DOI:** 10.3390/diagnostics14161750

**Published:** 2024-08-12

**Authors:** Giannapia Affaitati, Raffaele Costantini, Michele Fiordaliso, Maria Adele Giamberardino, Claudio Tana

**Affiliations:** 1Department of Innovative Technologies in Medicine and Dentistry, Center for Advanced Studies and Technology (CAST), G. D’Annunzio University of Chieti, 66100 Chieti, Italy; gpaffaitati@unich.it; 2G. D’Annunzio University of Chieti, 66100 Chieti, Italy; raffaele.costantini@unich.it; 3Department of Medicine and Ageing Sciences, G D’Annunzio University of Chieti, 66100 Chieti, Italy; michele.fiordaliso@gmail.com; 4Headache Center, Geriatrics Clinic, Department of Medicine and Science of Aging, Center for Advanced Studies and Technology (CAST), G. D’Annunzio University of Chieti, 66100 Chieti, Italy; mag@unich.it; 5Headache Center, Geriatrics Clinic, SS Annunziata Hospital, 66100 Chieti, Italy

**Keywords:** visceral pain, migraine, tension-type headache, irritable bowel syndrome, dysmenorrhea, endometriosis, myocardial infarction, angina, chronic pelvic pain, painful bladder syndrome

## Abstract

Headache and visceral pain are common clinical painful conditions, which often co-exist in the same patients. Numbers relative to their co-occurrence suggest possible common pathophysiological mechanisms. The aim of the present narrative review is to describe the most frequent headache and visceral pain associations and to discuss the possible underlying mechanisms of the associations and their diagnostic and therapeutic implications based on the most recent evidence from the international literature. The conditions addressed are as follows: visceral pain from the cardiovascular, gastrointestinal, and urogenital areas and primary headache conditions such as migraine and tension-type headache. The most frequent comorbidities involve the following: cardiac ischemic pain and migraine (possible shared mechanism of endothelial dysfunction, oxidative stress, and genetic and hormonal factors), functional gastrointestinal disorders, particularly IBS and both migraine and tension-type headache, primary or secondary dysmenorrhea and migraine, and painful bladder syndrome and headache (possible shared mechanisms of peripheral and central sensitization processes). The data also show that the various visceral pain–headache associations are characterized by more than a simple sum of symptoms from each condition but often involve complex interactions with the frequent enhancement of symptoms from both, which is crucial for diagnostic and treatment purposes.

## 1. Introduction

Pain from internal organs is amongst the most frequent pathologic conditions in internal medicine, representing one of the major causes of requests for medical consultation [1]. Headache, a very common and disabling neurological disorder, especially in its primary forms of migraine and tension-type headache is highly comorbid with a number of other pain conditions, such as fibromyalgia, myofascial pain syndromes, and, mostly, visceral pain [2,3,4]. Internists are thus very often faced with complex clinical pictures in patients with both headache and pain from internal organs, which pose both diagnostic and therapeutic challenges. Examples of visceral pain conditions highly comorbid with headache are numerous, including thoracic organs (i.e., heart: ischemic heart disease), abdominal organs (i.e., intestine: irritable bowel syndrome), and pelvic organs (i.e., female reproductive organs: primary dysmenorrhea, chronic pelvic pain from endometriosis; urinary tract organs: painful bladder syndrome), only to mention some [5,6,7,8,9,10,11,12,13,14,15,16,17,18,19,20,21,22,23,24,25,26].

The main question arising in this comorbidity context regards the possible link between visceral pain and headache; are concurrent headache and visceral pain mere associations or do they share some common pathophysiological aspect? Another question regards the overall clinical presentation of the comorbid patient; is it the mere sum of the single pain conditions or is there a mutual influence of symptoms between the two pain conditions? These questions could have important implications/repercussions for both the diagnosis and management of the comorbidity-affected patient. After an indispensable premise on visceral pain features and mechanisms, the present narrative review examines the main patterns of visceral pain–headache associations, their possible common pathophysiological aspects, the clinical presentation of the comorbid conditions, and the relative therapeutic implications.

## 2. Pain from Internal Organs: Characteristics and Main Mechanisms

Visceral pain is experienced by the vast majority of the population at some point in their life, whether in acute, recurrent, or chronic form. It can regard thoracic, abdominal, and pelvic internal organs, sustained by both organic and functional mechanisms. Independently of the primarily involved organ, visceral pain is expressed clinically with typical features, and it is generally time-dependent, i.e., is has different characteristics in the early vs the subsequent phases of a noxious visceral process [27,28,29,30,31,32].

True visceral pain is the sensation arising from internal organs in the early phases; it is described as a dull, vague, poorly discriminated sensation along the midline of the thorax or abdomen, anteriorly or posteriorly in the interscapular region, a common site independent of the affected organ. Its intensity is very variable, from low to high; however, it is not correlated with the extent of the internal damage, i.e., a mild pain can correspond to severe internal damage, as in the case of a silent myocardial infarction, while a high pain intensity can be expressed by dysfunctional conditions which are not life-threatening. The pain is always accompanied by marked neurovegetative signs, such as nausea, vomiting, profuse sweating caused by changes in heart rate, frequency of urination, and alvus disturbances. Emotional reactions, such as anguish, anxiety, and a sense of impending death, are also frequently present. At times, neurovegetative symptoms and emotional reactions are the only symptoms present and “replace” the pain, which is rather described as a sense of malaise or oppression. These accompanying neurovegetative signs/symptoms are very similar to those typical of migraine attacks [27]. The reasons beyond the characteristics of true visceral pain lie in the pathophysiology of visceral pain transmission. Most internal organs have a bilateral afferent innervation, and this explains why the sensation is normally felt around the midline; visceral organs also have a low density of sensory innervation with an ample functional divergence of the visceral afferent input within the central nervous system, which explains the vague nature of the sensation, its poor localization, and its diffuse nature. In addition, a certain degree of viscero-visceral convergence of the afferent input from internal organs exists in the central nervous system (CNS), i.e., different organs project their sensory input onto common neurons. This phenomenon accounts for the common site of pain perception in the “true visceral pain” phase, whatever the affected organ. Subsequent to this phase, either in the same episode or in subsequent episodes from the same organ, visceral pain changes its characteristics, becoming more similar to pain directly originating from somatic structures. It becomes “referred”, i.e., perceived in areas other than those in which the primary noxious event has taken place. These areas now differ according to the organ in question, generally being the somatic zones with the same sensory innervation as the affected organ. Examples are cardiac pain perceived at the level of the chest, left part (precordial area), or left shoulder/arm/forearm, sometimes irradiated to the neck and jaw, or pain from the biliary tree, referred to the right upper abdominal quadrant, with frequent radiation to the back, towards the angle of the scapula, or pain from the upper urinary tract, referred to the lumbar region ipsilateral to the affected kidney/ureter [33,34,35,36]. In the referred phase, the sensation becomes better localized and defined, neurovegetative signs are less marked, and emotional signs disappear. Referred pain from viscera can be categorized into two different entities: “referred pain without hyperalgesia” and “referred pain with hyperalgesia”. The former is normally the first to appear after the phase of true visceral pain. Also called “segmental pain”, it is not accompanied by any sensory change in the somatic area of referral, i.e., additional stimuli applied onto the area of referral do not increase the pain symptoms. Referred pain with hyperalgesia tends to occur later in the course of the visceral noxious process; here again, this can happen during the same episode of referred pain, if this is particularly prolonged, or in subsequent pain episodes in the case of recurrent visceral pain. Here, the sensation is accompanied by sensory changes in the somatic tissues of the body wall; additional stimuli applied onto these tissues increase pain symptoms, and the pain threshold (minimal intensity of a stimulus corresponding to the first report of pain by the patient) is decreased. Hyperalgesia in referred visceral pain areas first appears in the muscle layer, often accompanied by a state of sustained muscle contraction, but it can also extend to more superficial layers, i.e., subcutaneous tissue and skin, especially in cases of particularly prolonged and/or repetitive visceral pain episodes, as testified by decreased pain thresholds to different stimuli, e.g., electrical, according to a methodology well standardized in previous studies [31,32,33]. Numerous studies in patients affected with different pain conditions of internal organs have documented the relationship between the visceral pain expression from a specific organ and the extent and duration of the referred hyperalgesia. Muscle hyperalgesia occurs relatively early in the course of a visceral pain process, is accentuated in extent by the repetition of the episodes (e.g., repeated colics), and is long-lasting, persisting long after the spontaneous pain has ceased or often even after the healing—or apparent healing—of the involved organ. For instance, patients who have suffered from urinary colics from calculosis have an accentuation of the referred lumbar muscle hyperalgesia (i.e., greater threshold lowering) with the repetition of the colics, and they maintain this hyperalgesic status even in the pain-free interval. Lumbar muscle hyperalgesia often remains detectable years after the last pain episode, even after expulsion of the stone [31]. Areas of pain referral from viscera, where hyperalgesia is present, are also the site of trophic changes in deep somatic tissues, i.e., reduced thickness and section area of the muscle and increased thickness of the subcutaneous tissue [32,37].

The phenomenon of referred pain without hyperalgesia is attributed to the convergence of visceral and somatic afferent fibers onto the same neurons in the central nervous system (convergence–projection theory). Neurons receiving visceral afferent input also systematically receive afferent fibers from somatic structures, but the individual would attribute a sensation originating in the organ to the somatic structures due to mnemonic traces of previous experiences of somatic pain, more numerous than those of visceral pain in the course of life. In the case of referred pain with hyperalgesia, it is hypothesized that viscero-somatic convergent neurons become sensitized due to the nociceptive input from the internal organ; this facilitates the central processing of even normal inputs coming from the somatic area of referral, accounting for the hyperalgesia (convergence–facilitation). Pure central mechanisms, however, are not able to explain all the changes in the area of referral, particularly the trophic changes. These have instead been attributed to the activation of reflex arcs from the affected organ. The nociceptive input from the viscus would thus not only sensitize sensory neurons but also reflexively excite motoneurons towards muscles of the referral areas, generating a sustained contraction which would be responsible for further sensitization (via excitation of nociceptors in the muscle) and for the initiation of a dystrophic reaction. A reflex would also be activated, in which the efferent branch would be sympathetic towards skin/subcutis, being responsible for the increased subcutis thickness detected in the referred pain area [38,39,40,41,42].

While most visceral pain conditions, such as urinary or biliary colics, present hyperalgesia only in the district of pain referral, with sensory normality in control areas, other painful visceral diseases are characterized by the presence of generalized hypersensitivity to pain stimuli, as testified by a diffuse (in painful and nonpainful areas) decrease in pain threshold to different stimuli at the somatic level, especially in the muscle layer, an index of a central sensitization process [43,44,45]. Conditions such as irritable bowel syndrome, primary and secondary dysmenorrhea, chronic pelvic pain in endometriosis, or painful bladder syndrome are demonstrative examples of visceral pains in the context of a central sensitization state [32,37,46]. Interestingly, these same conditions are frequent comorbidities of other pain diseases, such as fibromyalgia or headache at a high frequency of attacks/chronic, which are recognized to be associated with generalized hyperalgesia [5,37,44].

In the next sections, specific associations of visceral pains and headache forms will be dealt with from diagnostic, pathophysiological, and therapeutic aspects (Appendix A).

## 3. Pain from Thoracic Internal Organs and Headache

### Cardiac Pain Plus Headache

Cardiac pain is typically expressed in most cases of coronary artery disease (CAD) [47,48,49]. CAD is very frequent in the general population and represents the single highest cause of death in the United States. Until the age of 55 years, it prevails in men due to the protective action of female sex hormones against atherosclerosis during the fertile years, but after menopause, it increases progressively in women to reach an equal distribution in the two sexes after the age of 65 years [48,50,51,52,53]. Mortality rates in men are 4-fold higher than those of women before the age of 55, see [24].

The main adequate stimulus for cardiac pain is ischemia, resulting from coronary stenosis and/or coronary spasm, which excites chemosensitive and mechanoreceptive receptors in the heart. Ischemic episodes release modulators including adenosine and bradykinin, that excite the receptors of the sympathetic and vagal afferent pathways. Cell bodies of the sympathetic afferent fibers from the heart and coronary arteries are concentrated in the dorsal root ganglia of the T2-T6 spinal segments, but they can spread as far as the C8-T9 segments. In the spinal cord, sympathetic afferent fibers from the heart synapse on cells of origin of ascending pathways. The spinothalamic tract projects to the medial and lateral thalamus and activates several cortical areas, including the anterior cingulate gyrus, the lateral basal frontal cortex, and the mesiofrontal cortex [54]. 

Spinal neurotransmission involves substance P, glutamate, and transient receptor potential vanilloid-1 (TRPV1) receptors; the release of neurokinins such as nuclear factor kappa b (NF-kb) in the spinal cord can modulate neurotransmission. Vagal cardiac afferent fibers likely mediate atypical anginal pain and contribute to cardiac ischemia without accompanying pain via relays through the nucleus of the solitary tract and the C1-C2 spinal segments [55,56].

The association between cardiac pain from CAD and headache, particularly migraine, has been extensively investigated by many authors in the course of several decades, although the outcome of the studies has not always proven univocal. Overall, migraine has been associated with cardiovascular disease (CVD) events among middle-aged adults; in particular, people with migraine with aura have been shown to present an increased risk of myocardial infarction, atrial fibrillation, and cardiovascular death compared with those without migraine [57,58].

In a prospective cohort study by Kurth et al. [16], 27,840 women from the United States were examined who were aged ≥45 years and suffered from migraine but not from cardiovascular disease (CVD) and angina at study start. In a mean 10-year follow-up, 580 major cardiovascular events were recorded in this sample. While migraine without aura was not linked to an increased cardiovascular risk, migraine with aura appeared significantly associated with this risk, especially myocardial infarction and angina, with respect to non-migraine women. Schürks et al. [59] instead failed to show any association between migraine and coronary heart disease or myocardial infarction, differently from previously reported data.

A prospective cohort study by Kurth et al. [15] assessed men aged 40–84 years not affected with CVD at study entry. Over a follow-up period of 15.7 years, 1449 major CVD events, particularly myocardial infarction and angina, occurred in migraine sufferers, significantly more than in non-migraine sufferers.

A post hoc subgroup analysis of the Women’s Health Study by Kurth et al. [14] randomized a 100 mg dose of aspirin on alternate days in primary prevention of CVD among 39,876 women aged ≥45 years. Women affected with migraine with aura, assigned to aspirin, nonsmokers, and affected with hypertension presented a significant increase in the risk of myocardial infarction.

A meta-analysis of previous publications (up to April 2014), published by Sacco et al. in 2015 [21] addressed in particular the relationship between migraine and ischemic heart disease. An increased risk of myocardial infarction and angina was found in migraineurs vs. non migraineurs, which was statistically significant.

In a review paper published in 2021, Gupta and Srivastava [60] reported that although the literature shows a higher prevalence of obesity, dyslipidemia, and family history of cardiovascular diseases in migraineurs, the ‘migraine–vascular’ connection persists in models where the traditional vascular risk factors are adjusted, implicating a migraine-specific pathophysiology at work. There is indeed some evidence linking an adverse vascular risk factor profile to incident myocardial infarction in migraineurs. 

A review of the subsequent year by Ng et al. [61] examined electronic databases (PubMed, Embase, and Scopus) until 22 May 2021 for prospective cohort studies evaluating the risk of myocardial infarction, stroke, and cardiovascular mortality in migraine patients. A random effects meta-analysis model was used to summarize the selected studies, which included 18 prospective cohort studies including 370,050 migraine patients and 1,387,539 controls. The result of the analysis showed that migraine, particularly migraine with aura, is associated with myocardial infarction and ischemic and hemorrhagic stroke. Migraine with aura increases the risk of overall cardiovascular mortality.

Fuglsang et al. [62] examined the impact of migraine on the risk of premature (age ≤ 60 years) myocardial infarction and ischemic/hemorrhagic stroke among men and women. Using Danish medical registries, a nationwide population-based cohort study was conducted between 1996 and 2018 among individuals aged 18–60 years with a median age of 41.5 years for women and 40.3 years for men. The results of the study made the authors conclude that for premature myocardial infarction and hemorrhagic stroke, there may be an increased risk associated with migraine only among women.

The pathophysiologic basis of the association between cardiac ischemic pain and migraine remains to be established. One possible underlying mechanism is based on the higher levels of certain serum markers in migraineurs vs. controls, namely, the pro-brain natriuretic peptide (pro-BNP), which suggests a preclinical cardiac involvement in the patients, and pro-inflammatory substances such as IL-1beta and IL-6 [63]. 

A second possible mechanism is related to the significant reduction in the number and activity of the circulating endothelial progenitor cells (EPCs) demonstrated in migraine patients, particularly those affected with migraine with aura vs. patients with tension-type headache and controls [64,65]. This hypothesis is based on the role played by EPCs in the repairing process and angiogenesis of ischemic tissues; they are responsible for neo-angiogenesis after ischemic insult. Deficiency or loss of EPCs can interrupt the balance between endothelial damage and restorative function, leading to endothelial dysfunction and atherosclerosis, thus promoting an increased risk of cardiovascular events in general, including cardiac ischemic pain.

In line with this hypothesis are also the results of the study by Rodríguez-Osorio et al. [66]. The authors evaluated flow-mediated dilation (FMD) in the dominant brachial artery and levels of calcitonin gene-related peptide (CGRP), vascular endothelial growth factor (VEGF), nitric oxide stable metabolites (NOx), and EPCs in the peripheral blood in patients with episodic migraine as compared to controls. The evaluation was carried out during the interictal periods and during the migraine attacks. Migraineurs presented significantly lower levels of EPCs, with EPC counts decreasing as migraines progressed in time. Migraineurs also had higher levels of CGRP, VEGF, and NOx, especially during the pain attacks.

Oxidative stress is another important factor that has been claimed to play a role in the link between cardiovascular risk and migraine in terms of altered oxidizing compounds and antioxidant processes. Oxidative stress has indeed been considered as an important factor in the pathogenesis of migraine headaches and can cause alterations in the cerebral blood flow [67].

Furthermore, several studies have addressed the possible role of cholesterol levels in the association of migraine and CV risk. The HUNT study [68], for instance, performed on over 48,000 subjects aged ≥20 years, revealed an elevated cardiovascular risk profile, as assessed through calculation of the Framingham 10-year risk score for coronary death and myocardial infarction in migraine, both with and without aura, and in non-migrainous headache, with the highest risk being in migraine with aura (the Framingham risk score assesses the risk of cardiovascular events considering multiple cardiovascular risk factors, including cholesterol levels, in combination to provide a single quantitative estimate of risk) [69]. The increase in migraine frequency was correlated to an increase in the Framingham score.

Patients with migraine, particularly those with aura, were also found to be more likely to receive a diagnosis of hypercholesterolemia than control subjects in a population-based study by Bigal et al. [70] and in the cross-sectional Epidemiology of Vascular Ageing Study [71], which also documented elevated values of triglycerides.

Tana et al. [72] showed a direct relationship between cholesterol levels, both total and LDL, and migraine frequency and intensity. Their study also showed that an effective migraine-preventative therapy was able to specifically impact cholesterol, determining a significant reduction in its levels. Though of a retrospective nature and performed on a relatively small sample of subjects, the study highlights the importance of a correct migraine prevention for better metabolic lipid control and consequently also for the possible modulation of the cardiovascular risk and the visceral pain conditions originating from ischemia due to atherosclerosis.

A link between migraine without aura and cardiovascular risk is also highlighted by microRNAs studies (miRNAs are short, non-coding RNAs which are de-regulated in various pathologic conditions). Female migraine patients without aura vs. controls show an altered expression of four miRNAs: miR-27b is up-regulated, while miR-181a, let-7b, and miR-22 are down-regulated, the same miRNAs known to be deregulated in atherosclerosis, see [28].

However, the role of an increased risk of atherosclerosis in migraine as a promoter of an increased risk for ischemic heart disease remains controversial, and various authors have failed to show a specific link between the two risk profiles. A cross-sectional case–control study by Stam et al. [73] specifically investigated if an enhanced risk of atherosclerosis in migraineurs represents the basis for the increased risk for ischemic stroke and coronary artery disease in this condition. Participants from the population-based Erasmus Rucphen Family study were studied. Atherosclerosis was assessed (intima media thickness, pulse wave velocity, and ankle–brachial index) in 360 migraineurs (209 without aura and 151 with aura) and 617 subjects without migraine or severe headache. Migraineurs, especially with aura, were found to be more likely to smoke, have diabetes, or have a modestly decreased HDL-cholesterol, while no differences were found for the atherosclerosis parameters. On this basis, the authors concluded that patients affected with migraine have no increased risk of atherosclerosis and that therefore enhanced atherosclerosis does not represent the pathophysiological basis for the increased cardiovascular risk of migraineurs. In a very recent study by van Welie et al. [74], sex-specific metabolomics were analyzed to identify markers that may explain the migraine–cardiovascular disease (CVD) relationship. Plasma metabolome analyses showed alterations in migraine patients. Sex-specific findings showed a less CVD-protective HDL metabolism as well as the ApoA1 lipoprotein, especially for women with migraine. The authors found that there was no general large dyslipidemia profile in migraine patients, in line with the already reported findings that the increased risk of CVD in migraine patients seems not to be due to (large-artery) atherosclerosis. Sex-specific associations indicate a less CVD-protective lipoprotein profile in women with migraine. 

Clearly, more studies are needed to better define the possible relationship between increased atherosclerosis risk in migraine and increased ischemic heart disease in the migraine population.

In synthesis, there is an association between migraine and ischemic heart disease (IHD) [75], and migraine with aura can be considered one of the risk factors for IHD. Gender differences in the association between ischemic heart disease and migraine are not definitely clarified due to the lack of systematic studies examining men and women separately with respect to the comorbidity. The studies so-far available, however, would point to a higher rate of the association in women than in men throughout the life span, irrespective of the different frequency of occurrence of ischemic heart disease in the fertile vs. postmenopausal phase in women. Although the exact relationship between migraine and ischemic heart disease is still not completely known, some studies suggest the association to be multi-factorial. In addition to the already-reported mechanisms of endothelial dysfunction and oxidative stress, a genetic basis for the association has also been put forward; if migraine and coronary artery disease share the same pathological process, they likely share genetic risk as well. Genetic studies, however, have shown controversial results; Winsvold et al. [76] determined the genetic overlap between migraine and CAD by performing analyses based on large genome-wide association study (GWAS) meta-analyses of migraine (19,981 cases, 56,667 controls) and coronary artery disease (21,076 cases, 63,014 controls). They found a significant overlap of genetic risk loci for migraine and CAD. One shared risk locus was identified as the encoding phosphatase and actin regulator 1 (PHACTR1) gene. This is a genome-wide significant risk locus for both migraine and CAD. This protein phosphatase 1 binding protein is highly expressed in the brain, where it regulates synaptic activity and dendritic morphology. It is also expressed in arteries and plays an important role in the regulation of endothelial function and is associated with altered vasomotor tone. When stratified by migraine subtype, this was limited to migraine without aura, and the overlap was protective, in that patients with migraine had a lower load of CAD risk alleles than controls. Thus, the authors concluded that shared biological processes probably contribute to the risk of migraine and CAD, but surprisingly, this commonality is restricted to migraine without aura, and the impact is in opposite directions. 

Another possible underlying common factor between migraine and ischemic heart disease is represented by hormonal factors. Sex hormones seem to be related to migraine, and this may be responsible for the gender difference seen in migraine patients. Fluctuating levels of sex hormones, especially estrogen, predispose females to migraine attacks by increasing cortical excitability. Estrogen has a major influence on vascular health, as it is involved in both thrombotic as well as vasodilatory mechanisms. However, the role of estrogen in causing IHD in migraine patients remains unclear because of the complex mechanisms involved [77].

Independently of the exact mechanisms underlying the comorbidity between migraine and ischemic heart disease, however, overall, the outcome of the reported studies on the link between cardiovascular risk and migraine suggests important repercussions on the therapeutic approach to patients with these conditions.

Given the link between migraine and increased risk for cardiovascular events, caution should be placed in migraineurs in employing treating drugs, such as triptans or Non Steroidal Anti-inflammatory Drugs (NSAIDs). The latter, in addition to their gastrointestinal and renal side effects, notably have an impact on the CV system [78,79]. This class of compounds, particularly important for extinguishing acute migraine crises, need to be prescribed after an accurate evaluation of the risk profile of the patients with respect to the CV event as well as in relation to the presence of other CV risk factors, such as hypertension, obesity, or smoking habits [24]. The importance of preventing chronicization of migraine through prophylaxis also emerges in this context, not only to avoid the burden of the increased pain load to the patients due to the condition but also to better control the cardiovascular risk involved in increasing symptomatic drug consumption. 

Prevention of chronicization is also important in tension-type headache (TTH), though specific data on the link of this type of headache and CV risk remains to be demonstrated. Since NSAIDs are the first-line therapy for TTH, it is important to limit their use/prevent their abuse to avoid an increased CV risk linked to the use of these drugs [78].

In the context of headache–cardiovascular pain comorbidity, it is also important to consider the possible CV risk profile of the new treatments for migraine (e.g., ditans, gepants, mAbs). Robblee and Harvey [80] present data on cardiovascular safety for the new acute and preventive migraine treatments, including ditans, gepants, and calcitonin gene-related peptide monoclonal antibodies (CGRP mAbs) alongside older medications like triptans and ergotamines. The authors conclude that there are no cardiovascular safety concerns for lasmiditan, and triptans are safer in cardiovascular disease than their contraindications may suggest. There is insufficient evidence that gepants and CGRP mAbs should be contraindicated in those with cardiovascular disease. 

## 4. Pain from Abdominal Internal Organs and Headache

### 4.1. Pain from the Gastrointestinal Tract Plus Headache

Gastrointestinal symptoms and headache are strictly related. Nausea and vomiting are frequently present during a migraine attack, and gastrointestinal pain/disturbance can be part of the range of headache conditions, as classified by the International Headache Society [3].

#### 4.1.1. Visceral Symptoms Associated with Headache

Episodic syndromes that may be associated with migraine (code 1.6) is the label under which headache conditions are listed in the Headache Classification, which are expressed clinically with significant gastrointestinal symptoms. Recurrent gastrointestinal disturbance (code 1.6.1), previously named chronic abdominal pain; functional abdominal pain; functional dyspepsia; irritable bowel syndrome; or functional abdominal pain syndrome are described as recurrent attacks of abdominal pain and/or discomfort, nausea, and/or vomiting, manifesting infrequently, chronically, or at predictable intervals, that may be associated with migraine. The diagnosis requires the occurrence of at least five attacks (of abdominal pain and/or discomfort and/or nausea and/or vomiting) in the absence of any abnormality at gastrointestinal examination and evaluation alongside an exclusion of any other disorder which can motivate the symptoms. Subcategories are cyclical vomiting syndrome (code 1.6.1.1) and abdominal migraine (code 1.6.1.2).

Cyclic vomiting syndrome is described as “Recurrent episodic attacks of intense nausea and vomiting, usually stereotypical in the individual and with predictable timing of episodes. Attacks may be associated with pallor and lethargy. There is complete resolution of symptoms between attacks”. The attacks, which are very disabling, have a predictable cyclic nature, normally starting at the same time of day, with a similar profile of symptom presentation: a similar duration and intensity as well as a similar type of associated signs. Prodromal phenomena often occur, with triggering factors frequently identifiable, such as lack of sleep, certain foods, menstruation, stress, and physical effort. Given the similarity of the characteristics of this syndrome to those of migraine, it has traditionally been suggested that cyclic vomiting syndrome is indeed a condition related to migraine. Although a typical occurrence in childhood, the syndrome can also manifest in the adult population, with its prevalence still remaining a matter of debate [81,82,83,84]. An analysis of the relative literature indeed shows that CVS can be regarded as a disorder of the migraine spectrum due to the high degree of its comorbidity with migraine in the adults but also its responsiveness to various preventative migraine agents and also to injectable sumatriptan in some cases, even in the absence of headache [3,85].

At functional magnetic resonance imaging, both CVS and migraine patients showed diminished insular connectivity with the sensorimotor network, suggesting a common pathophysiology [82].

Abdominal migraine is another subcategory of recurrent gastrointestinal disturbance in the classification (code 1.6.1.2). It affects from 0.2% to 4.1% of children and consists of paroxysmal, recurrent, and acute abdominal pain attacks (midline abdominal pain of moderate–severe intensity) of a 2–72 h duration with associated symptoms, including pallor, nausea, vomiting, and vasomotor symptoms. In-between episodes, patients return to their baseline health. 

It is a disorder of unknown origin, but hypothesized contributors to its pathophysiology include a combination of visceral hypersensitivity, gut–brain enteric nervous system alterations, and psychological factors. Treatment mainly consists of preventive measures, most of which are nonpharmacologic. Possible pharmacologic treatments include abortive drugs used for migraine attacks such as analgesics and antiemetics. Indeed, though headache does not occur during episodes of abdominal migraine, most affected children will develop migraine later in life, which stresses the importance of a careful collection of the clinical history in adult headache patients, asking specifically not only for the presence of head pain in childhood but also for the occurrence of migraine equivalents, such as abdominal pain with the characteristics of abdominal migraine [3,86,87].

#### 4.1.2. Functional Gastrointestinal Diseases and Headache

Functional gastrointestinal disorders (FGIDs) are common disorders involving persistent and recurring GI symptoms, which occur as a result of abnormal functioning of the GI tract. Among FGIDs, irritable bowel syndrome (IBS) is the most prominent. 

IBS is characterized by recurrent episodes of abdominal pain or discomfort and changes in bowel habits, such as constipation, diarrhea, or both, in the absence of documentable detectable organic abnormalities. Diagnosis is performed through the Rome IV criteria [four subtypes: predominant diarrhea [IBS-D], predominant constipation [IBS-C], mixed bowel habits [IBS-M], unclassified IBS [IBS-U]]. The worldwide prevalence of IBS is around 11%, and women are more frequently affected (1.8–2:1 ratio) and also have a tendency to suffer from more intense symptoms than men, in particular abdominal pain and constipation-related symptoms. The pathophysiology of IBS is multifactorial and yet not fully elucidated. A key role is believed to be played by disturbances in the brain–gut axis (BGA). The BGA includes the enteric nervous system (ENS) and gut wall peripherally, the central nervous system (CNS), and the hypothalamic–pituitary–adrenal (HPA) axis. There are bi-directional interactions between the gut and CNS, involving neural, endocrine, and neuroimmune pathways; in normal circumstances, signals from the GI tract influence the brain, which in turn influences in motility, secretion, and immune function. Disruptions of this system contribute to IBS occurrence through a complex combination of factors—from genetic to environmental. In IBS, changes in intestinal motility have been attributed to disruptions in the metabolism of serotonin (5-HT), a key neurotransmitter within the BGA, released by enterochromaffin cells in the ENS, whose main functions are the stimulation of gut peristalsis and the regulation of secretory and vasodilator activities. Molecular biology studies have also demonstrated the presence of low-grade mucosal inflammation and immune dysfunction in IBS. Mucosal inflammation promotes visceral hypersensitivity in the GI tract due to peripheral sensitization (enhanced perception of pain in response to GI stimuli). Enhanced nociceptive barrage from the periphery secondarily promotes phenomena of central sensitization, which further contribute to GI pain symptoms/sensitivity. Gut dysbiosis, an imbalance in gut microbiota due to several internal and external factors, also significantly contributes to IBS. 

Dysbiosis of the gut microbiota contributes to promoting several inflammatory and immunological modifications, which can undermine the structure of the GI mucosal barrier through an increase in intestinal permeability, consequently perturbing the state of homeostasis within the GI tract. Disruption in barrier integrity is associated with abdominal pain and visceral sensitivity in IBS and exposes neural and immune components to luminal microbes [88].

Numerous studies have established the high degree of comorbidity between functional gastrointestinal disorders (FGIDs) and headache, especially migraine, also including medication-overuse-headache [5,8,17,18,25,89,90,91,92,93].

Subjects complaining of reflux symptoms, nausea, diarrhea, or constipation were indeed shown to present a higher prevalence of headache, including migraine, with respect to subjects not complaining of these same symptoms, as shown by a population-based, cross-sectional study carried out by Aamodt et al. [5]. 

Patients affected with irritable bowel syndrome (IBS) vs. healthy subjects displayed a higher prevalence of migraine in a cohort study of almost 98,000 IBS patients (6% vs. 2.2%) [91].The study by Meucci et al. [18], instead, represents an exception, since it did not demonstrate any difference in the prevalence of migraine in patients affected with reflux-like/ulcer-like dyspepsia vs. healthy subjects evaluated in a cohort of dyspeptic patients referred for upper-gastrointestinal endoscopy. A subsequent study by Lackner et al. [93] in 175 IBS patients referred to two specialty care clinics showed an average of five comorbidities in IBS, among which tension-type headache was among the most frequent; patients with tension-type headache plus IBS also displayed more severe IBS symptoms. Another study by Park et al. [94] exploring the presence of functional symptoms (FGID) in 109 migraineurs attending a teaching hospital in Korea showed that FGID symptoms are highly prevalent in migraine.

The pathophysiology of the frequent co-occurrence of functional gastrointestinal disorders and migraine is still a matter of investigation, and a complex combination of factors are probably involved in this framework, from genetic predisposition (migraine and IBS have a strong familiar aggregation) to neuroimmunity, neuroendocrine interactions, the role played by the brain–gut axis, as well as altered serotonin signaling [95,96,97,98]. Regarding genetic factors, it is worth noting that polymorphisms in the promoter region of the serotonin reuptake transporter (SERT) gene (SERT deletion/deletion genotype) are associated with IBS, particularly diarrhea-predominant [99], and the SERT gene polymorphism of the variable number of tandem repeats is associated with migraine [29,100]. Among the various pathophysiological hypotheses behind the comorbidity between functional gastrointestinal disorder and headache, however, an important role is attributed to mechanisms of central sensitization. As already stated above, in IBS, peripheral sensitization of visceral afferent fibers is documented, which is then followed by central sensitization secondary to the augmented sensory inflow to the central nervous system [101,102].

Clinically, IBS patients do show signs of sensitization, in that most of them present a diffuse decrease in the pain threshold of somatic tissues, similar to what is observed in migraine and tension-type headache patients with a high number of attacks or that are chronic [103]. Clinical studies also show that in IBS–headache co-occurrence, patients show enhanced headache symptoms due to the abdominal pain episodes, with headache attacks being triggered by the visceral pain episodes [6]. This phenomenon probably occurs as a consequence of the increased excitability of the CNS due to the nociceptive inputs from the GI tract, with spreading to the trigeminal system and facilitation of the transmission of headache pain.

The co-occurrence of headache and functional gastrointestinal disorders has important implications for therapy. There are common pharmacologic and non-pharmacologic measures that can be used in the two sets of conditions. The former include tricyclic antidepressants, serotonin noradrenergic reuptake inhibitors, and gabapentinoids, the same class of compounds that can be employed for preventative treatment in headache. Among the latter, psychological treatments, such as cognitive behavioral therapy, are effective for both FGIDs and headache. Even in the case of therapies useful for one condition only, e.g., specific diet regimens targeted to IBS, there is evidence of an indirect beneficial effect for headache, e.g., IBS patients with improved abdominal pain symptoms after an appropriate diet also show an improvement in headache symptoms with a reduction in both the number and intensity of headache attacks, probably as a consequence of the reduction in the central sensitization level due to the nociceptive inputs from the GI tract. All these observations point to the importance of an integrated approach to the gastrointestinal and headache conditions in comorbid patients [27]. 

The relationship between irritable bowel syndrome and migraine was also explored by Chang and Lu [104], highlighting the main similarities between the two conditions (from female predominance to the frequent association of both conditions with psychiatric comorbidities, fibromyalgia, chronic fatigue syndrome, and interstitial cystitis/chronic bladder syndrome, only to mention some). On this basis, the authors stressed a possible common origin for the two conditions, most probably mechanisms of central sensitization.

Di Stefano et al. [105] examined the relationship between migraine and functional dyspepsia by assessing if alterations to gastric sensorimotor activity may be related to migraine. They studied 60 patients affected with functional dyspepsia (38 with postprandial distress syndrome—PPDS—and 22 with epigastric pain syndrome). The results showed that 54% of the patients with epigastric pain syndrome had migraine, which was never correlated with meal ingestion. A total of 76% of the patients with postprandial distress syndrome suffered from migraine, 89% of whom had a migraine onset related to meals; the severity of migraine was significantly correlated with postprandial modification of the gastric discomfort threshold. Patients with postprandial distress syndrome associated with moderate-to-severe migraine had significantly higher severity of fullness and early satiation than patients with PPDS with mild-or-absent migraine. The authors concluded that in patients with functional dyspepsia and postprandial symptoms, migraine is a very frequent comorbidity and is associated with an increased severity of fullness and early satiation and, probably, correlated with postprandial hypersensitivity.

An experimental study carried out in animals by Mohammadi et al. [106] investigated the role of CGRP in cross-organ sensitization; since CGRP is an important mediator of migraine attacks, the study is relevant for the understanding of possible underlying common mechanisms of visceral pain–headache comorbidity. Irritable bowel syndrome and bladder pain syndrome/interstitial cystitis (BPS/IC) are comorbid visceral pain disorders seen commonly in women, which can involve visceral organ cross-sensitization. The authors employed a monoclonal anti-CGRP F(ab’) in three rodent models to test the hypothesis that visceral organ cross-sensitization is mediated by abnormal CGRP signaling. Adult female rats underwent a transurethral infusion of protamine sulfate (PS) into the urinary bladder or an infusion into the colon of trinitrobenzene sulfonic acid (TNBS) to induce cross-organ sensitization. The visceromotor response (VMR) to colorectal distension (CRD) was used to assess colonic sensitivity. The frequency of abdominal withdrawal responses (AWR) to von Frey filaments applied to the suprapubic region was employed to evaluate bladder sensitivity. Quantification of transepithelial electrical resistance (TEER) was employed to measure PS- or TNBS-induced changes in colonic and bladder permeability.

Peripheral administration of an anti-CGRP F(ab’)_2:_ inhibited PS-induced visceral pain behaviors and colon hyperpermeability and decreased TNBS-induced pain behaviors and colon and bladder hyperpermeability. PS into the bladder or TNBS into the colon significantly increased the VMR to CRD and AWR to suprapubic stimulation and decreased bladder and colon TEER. According to the authors, these results suggest that peripheral CGRP plays and important role in visceral nociception and organ cross-sensitization, supporting the notion that CGRP could be a therapeutic target for visceral pain in patients with IBS and/or BPS/IC. Thus, since a monoclonal antibody against CGRP was found to reduce concomitant colonic and bladder hypersensitivity and hyperpermeability, CGRP-targeting antibodies, in addition to migraine prevention, may represent a novel treatment approach to multi-organ abdominopelvic pain following injury or inflammation. Furthermore, since monoclonal antibodies against CGRP are crucial in migraine treatment, CGRP activity (though mostly peripheral in visceral pain and central in migraine) could be a specific common pathogenetic link between IBS/PBS and migraine comorbidity.

#### 4.1.3. Organic Gastrointestinal Diseases and Headache

Another link between headache and GI pain disorders has been hypothesized, namely, between migraine and gallbladder disease. In a study by Chen et al. [107], migraine risk was assessed for patients with GSD, i.e., 20,427 patients diagnosed with GSD between 2000 and 2011 from Taiwan’s National Health Insurance Research Database (NHIRD). Controls (n. 81,706) were randomly selected from the non-GSD population with frequency matching by age and index year for the control cohort. All patient cases were followed until the end of 2011 to measure the incidence of migraines. The results showed that GSD is associated with an increased risk of migraines in the Taiwanese population, but the risk diminishes after a cholecystectomy. Furthermore, in the development of migraines, GSD is synergic with some migraine-associated comorbidities, such as CAD, depression, and anxiety.

Central sensitization processes, triggered by nociceptive inputs from the gallbladder, are likely to be underlying factors for the comorbidity.

### 4.2. Microbiota and Pain

The relationship between gut microbiota and pain, such as visceral pain, headaches (migraine), chronic abdominal pain (CAP), or joint pain, has received increasing attention in recent years [108,109,110].

In 2023, Liu et al. [111] published a review paper on the role of gut microbiota in chronic pain. Gut microbiota plays a pivotal role in modulating chronic pain; it is a key junction point between the neuroimmune–endocrine and the microbiome–gut–brain axes, potentially affecting chronic pain, either directly or indirectly. Various molecules, e.g., metabolites, neuromodulators, neuropeptides, and neurotransmitters, from the gut microbiota modulate phenomena of peripheral and central sensitization by targeting the relative receptors, thus affecting the progress of development of chronic pain. Dysbiosis of gut microbiota is furthermore associated with the evolution of different chronic pain conditions, such as visceral pain (the already-reported IBS) neuropathic pain, inflammatory pain, migraine, and fibromyalgia. This provides the rationale for prescribing probiotic supplementation in chronic pain, according to the authors. 

Another recently published study by He et al. [112] investigated the causal association between gut microbiome and migraine. In this study, the authors retrieved the single-nucleotide polymorphisms related to the gut microbiome from the gene-wide association study (GWAS) of the MiBioGen consortium. The summary statistics datasets of migraine were obtained from the GWAS meta-analysis of the International Headache Genetics Consortium (IHGC) and FinnGen consortium. 

In the IHGC datasets, ten, five, and nine bacterial taxa were found to have a causal association with migraine, MA (migraine without aura), and MO (migraine with aura), respectively. Genus Coprococcus3 and genus Anaerotruncus were validated in the FinnGen datasets. Nine, twelve, and seven bacterial entities were identified for migraine, MA, and MO, respectively. The causal association still existed in family Bifidobacteriaceae and order Bifidobacteriales for migraine and MO after FDR correction. According to the authors, these data show that gut microbiomes may exert causal effects on migraine, MA, and MO, providing novel evidence for the dysfunction of the gut–brain axis in migraine.

## 5. Pain from Pelvic Internal Organs and Headache

### 5.1. Female Reproductive Organs and Headache

Several visceral pain conditions originating from the female reproductive organs have been shown to be comorbid with headache [6,8,13,27,113,114,115,116]. 

#### Dysmenorrhea and Headache

Dysmenorrhea, i.e., painful menstruation, can be primary if there is no detectable organic disease in the female reproductive organs or secondary if an organic condition is ascertained, as in the case of endometriosis. Co-occurrence of dysmenorrhea, either primary or secondary, with headache is very common epidemiologically, particularly regarding migraine [6,117].

Primary dysmenorrhea and headache. Primary dysmenorrhea, i.e., cyclic pain at menses not due to pelvic organ disease or structural abnormality of the female reproductive area, affects over a half of women in their reproductive age. Due to an excess of prostaglandin production and increased myometrium contractility, it is typically cramp-like, perceived in the lowest abdomen, but it is often also in the lower back and upper thighs, frequently accompanied by neurovegetative signs such as nausea or vomiting, changes in heart rate, diarrhea, and emotional reactions [118,119]. Its typical time course involves starting a few hours before bleeding, with worsening during the menstrual flow and duration frequently for the whole period of menses.

As for all forms of visceral pain, primary dysmenorrhea is accompanied by secondary hyperalgesia in the areas of referred pain; this is most pronounced in the muscle tissue but can also extend to the overlying subcutis and skin in particularly severe cases. The muscle hyperalgesia accentuates with the repetition of the painful episodes, which, given the cyclic nature of the pain, corresponds to the duration of the disease, i.e., the higher the number of the years the patients have been suffering from dysmenorrhea, the more pronounced the muscle hyperalgesia is in the referred area. Though especially pronounced in the perimenstrual period, the hyperalgesia also persists in-between menses. Most women with primary dysmenorrhea also present some degree of muscle hyperalgesia in control areas, showing a tendency towards a diffusion of the phenomenon of somatic hypersensitivity, which typically characterizes central pain syndromes. Interestingly, this is a similar pattern observed in headache patients, both in migraine and tension-type headache, with a high number of crises. Indeed, primary dysmenorrhea and primary headache have a high degree of comorbidity [118]. Migraine very frequently occurs during menses, especially if these are painful. Although menstrual migraine, i.e., migraine exclusively present at menses, affects only 5–8% of women migraineurs, about 50% of all migraine women report an association of their migraine crises with the menstrual cycle [3]. A possible common mechanism between primary dysmenorrhea and migraine is represented by prostaglandin overproduction. Non-Steroidal Anti-Inflammatory Drugs (NSAIDs) are indeed effective in both conditions, representing the first-line treatment in dysmenorrhea but also an important symptomatic option for migraine when triptans either fail or are contraindicated for the specific patient [120,121]. Another common therapeutic measure is represented by hormone therapy, effective in most cases of primary dysmenorrhea and in menstrual migraine. Thus, the comorbidity between migraine and primary dysmenorrhea has important implications for treatment [6].

Comorbidities of both primary and secondary dysmenorrhea (see also section below) were examined in a study by Evans et al. [114]. Symptom questionnaires were completed by 168 women with dysmenorrhea, allocated to three groups based on their diagnostic status for endometriosis (confirmed, excluded, or unknown). Women with and without endometriosis had similar symptom profiles, with a mean of 8.5 symptoms each. Only the presence of stabbing pelvic pains was associated with more severe dysmenorrhea, more days per month of dysmenorrhea and of pelvic pain, and a diagnosis of migraine. 

Dysmenorrhea secondary to endometriosis and headache. 

Endometriosis affects around 10% of women in the reproductive phase of their life, reaching 25–35% in infertile women. The disease is defined as the presence of endometrial tissue in abnormal locations within the abdomino-pelvic cavity. The most frequently affected sites are the bowel surface, ovaries, uterine tubes, uterine ligaments, cul-de-sac, rectovaginal septum, cervix, and pelvic peritoneum. The main symptoms of endometriosis are subfertility or infertility, vaginal hyperalgesia, and dyschezia. Spontaneous pain, either as secondary dysmenorrhea or chronic pelvic pain, is a frequent, but not constant, occurrence in endometriosis patients; in women with chronic pelvic pain, active endometriosis is found in around 33% of cases. There is also no correlation between the extent of the endometriosis lesions and the occurrence and degree of the spontaneous pain, i.e., small lesions can often cause higher pain than larger and more diffuse lesions [122,123,124,125]. 

Women with pain from endometriosis also typically display referred somatic hyperalgesia at abdomen/pelvic level, similarly to women affected with primary dysmenorrhea. The hyperalgesia is prevalent at the muscle level, but it may also extend towards the surface or at the subcutis/skin level in particularly severe cases. There have been studies indicating that deep (muscle) somatic hyperalgesia is also present outside the areas of visceral pain referral, i.e., muscle hyperalgesia is diffuse, similarly to what is, again, observed in women with primary dysmenorrhea [32,126].

The mechanisms beyond endometriosis remain to be elucidated in full. Plausible hypotheses include retrograde menstruation, lympathic system spread, and hematogenous spread. Similarly, the pathophysiology of the pain in endometriosis is also incompletely known; possible mechanisms include excessive production of prostaglandins, increased peritoneal sensitivity, chemical peritoneal irritation, and bleeding at the site of endometriosis lesions [127,128].

Comorbidity of endometriosis with other chronic pain conditions is particularly high; it is estimated to be present in about 20% of endometriosis women, with the comorbid conditions including vulvodynia, chronic bladder pain syndrome, irritable bowel syndrome, fibromyalgia, and headache, particularly migraine [22,129,130,131,132].

Already in 1975, Tervila and Marttlia [133] suggested that endometriosis and headache were significantly associated; in a sample of 125 women subjected to surgical procedures because of pelvic pain, those with documented endometriosis complained of significantly more headaches perimenstrually than women with pelvic pain from causes other than endometriosis.

In 2004, Ferrero et al. [115] showed that endometriosis women vs. controls presented significantly higher levels of migraine, particularly migraine with aura, while the two groups did not differ regarding tension-type headache.

Three years later, a paper by Tietjen et al. [132] reported significantly higher percentages of endometriosis in migraineurs than controls, as evaluated in two groups of patients and healthy subjects, respectively, over a period of 2 years. Chronic headache was also significantly more frequent in comorbid patients (endometriosis + headache) than in patients with migraine only, with median scores of headache-related disability being significantly higher in the comorbid group. In addition, the endo + headache women also presented a higher level of other comorbid conditions, including visceral pain from organs other than those of the reproductive area, i.e., the gastrointestinal tract (irritable bowel syndrome) and the urinary tract (painful bladder syndrome), but also fibromyalgia, chronic fatigue syndrome, depression, and anxiety. Thus, from this study, it emerges that endometriosis has a higher prevalence in migraine women than in non-headache controls, with headache crises being more frequent and disabling in comorbid than non-comorbid patients, with it also being more frequently associated with other pain comorbidities in endo + migraine patients than in migraineurs without endometriosis.

In a recent review, Jenabi and Khazaei [116] examined the association between endometriosis and the risk of migraine; the authors evaluated published papers on the topic until May 2020 through international electronic bibliographic databases, including PubMed, Web of Science, and Scopus. The search identified 802 articles with 287,174 participants. A significant association was found between endometriosis and the risk of migraine headache.

Wu et al. [134] performed a case–control study of 167 endometriosis women and 190 control women (affected with other benign gynecological conditions) performed between September 2017 and January 2021. All women completed a self-administered headache questionnaire. Migraine was significantly more prevalent in endometriosis patients than in controls. For all endometriosis and control women, migraineurs were 4.6 times more likely to have severe endometriosis. The authors concluded that their study supports the strong association between migraine and endometriosis. 

A study by Gete et al. [12] examined the comorbidities and associated symptoms of endometriosis, including 7606 women born from 1973 to 1978 and using data from the Australian Longitudinal Study on Women’s Health that were collected every 3 years from 2009 to 2018. Women completed a checklist on the presence of 24 symptoms, among which was the presence of headache. A strong association was found, among other symptoms, with several forms of pain other than pelvic, i.e., backpain, painful joints, and headaches or migraines.

The basis of the frequent endometriosis–migraine comorbidity is likely to be multifactorial. A genetic common basis has been suggested by several studies, see [135]. In 2009, Nyholt et al. [19] hypothesized shared genes between the two conditions; other studies point to the estrogen receptor 1 gene as a common genetic factor between headache (the receptor has been implicated in migraine susceptibility) [136] and endometriosis (the same receptor has been associated with endometriosis) [137,138]. The ovarian hormonal asset is also likely to be crucial. Estrogen is reported to modulate visceral pain by enhancing neuronal activities or regulating neuronal plasticity processes at the level of the peripheral nervous system, spinal cord, and supraspinal nervous system [139].

Ovarian hormones influence both endometriosis and headache [140]. Migraine has maximal expression in women during their reproductive years; frequently, it starts with the first menstruation and improves, often ceases, after menopause. Early menarche is believed to represent a risk factor for both endometriosis and headache [113,141]; in addition, migraine seems to start earlier in women affected with endometriosis [115]. Possible mechanisms behind the higher prevalence of headache in women with early menarche are higher estrogen levels or increased estrogen sensitivity. Starting from the theory of retrograde menstrual blood in the case of endometriosis, provided this is true, one could affirm that all conditions leading to more menstruation (and this would be the case of early menarche) will trigger a higher prevalence of endometriosis. Along the line of a common hormonal basis, gonadotropin releasing hormone (GnRH) agonists (GnRH-a) have shown positive effects in the treatment of pain due to endometriosis, whilst at the same time, they also produced a decrease in migraine attacks [142,143]. These data stress the important role of the hormonal factor beyond both endometriosis and migraine, although the link appears more complex if we consider that headache is typically indicated as a side effect of GnRH-a treatment.

Increased plasma levels of prolactin have also been demonstrated in both endometriosis and migraine and can thus represent a further common underlying mechanism [144].

It has also been hypothesized that one condition causes the other. Endometriosis, similarly to chronic pelvic pain, is frequently associated with diffuse hyperalgesia of somatic tissues (decreased pain thresholds to different stimuli in both painful and nonpainful areas), the clinical correlation of a central sensitization state [27]. As reported by Raimondo et al. [145], central sensitization is indeed highly prevalent among women with endometriosis, particularly those presenting, among other factors, moderate–severe chronic pelvic pain, and three central sensitivity syndromes, i.e., migraine or tension-type headache, irritable bowel syndrome, and anxiety or panic attacks. It has therefore been suggested that nociceptive inputs from areas other than the sites of the endometriosis implants, including those from the cervicofacial district, are also facilitated. Thus, sensitization due to endometriosis would facilitate the occurrence of headache attacks [13].

A further hypothesis on the pathophysiology of the comorbidity is that the presence of migraine in a patient facilitates the diagnosis of endometriosis. Since migraine at a high frequency of attacks/chronic is characterized by diffuse somatic hypersensitivity, as testified by a generalized decrease in pain thresholds persisting beyond the headache attacks [146], migraine sufferers would be more susceptible than non-migraine subjects to suffer from pain of any origin, including the pain from endometriosis, a circumstance which would favor the diagnosis of endometriosis. This hypothesis needs confirmation with controlled studies.

Another link between the two conditions is that estroprogestins can be used in both. Estroprogestins relieve pain from endometriosis, but at the same time, they increase the risk of presenting endometriosis [147]. Estroprogestins and hormonal replacement therapy in general have also been associated with the occurrence of headache [148,149]. However, no studies so far have proven a direct causal link between an extensive use of estroprogestins and the occurrence of migraine and endometriosis; therefore, this hypothesis remains as such.

A further link between headache and endometriosis is represented by the common therapy with NSAIDs. Although treatment of endometriosis is complex and still unsatisfactory, NSAIDs are often used to relieve pain symptoms; they are also very often employed for the symptomatic treatment of headache, both migraine and tension-type headache [27].

Chronic pelvic pain and headache.

Chronic pelvic pain (CPP) is chronic or persistent pain perceived in structures related to the pelvis, whereby the perception indicates that the patient and clinician, to the best of their ability from the history, examination, and investigations (where appropriate) have localized the pain as being discerned in a specified anatomical pelvic area. It is often associated with negative cognitive, behavioral, sexual, and emotional consequences as well as with symptoms suggestive of lower urinary tract, sexual, bowel, pelvic floor, or gynecological dysfunction. In the case of documented nociceptive pain that becomes chronic/persistent through time, the pain must have been continuous or recurrent for at least three months [1]. It affects around 26% of the world’s female population, accounting for 40% of laparoscopies and 12% of hysterectomies. It is a complex disorder, which often overlaps with nonpelvic pain conditions, among which are fibromyalgia and migraine [10,150,151,152,153]. A possible common pathophysiological mechanism between chronic pelvic pain and migraine is represented by centrally mediated pain factors, i.e., central sensitization [45].

### 5.2. Urinary Tract Organs and Headache

Previously called interstitial cystitis, painful bladder syndrome (PBS) is a condition of chronic visceral pain, perceived at the supra-pubic level, accompanied by an array of urinary symptoms including urgency, nocturia, and urinary frequency. Its prevalence ranges from 2 to 17.3% in the general population (variability depending on the diagnostic criteria applied), increasing in the female sex and patients with one first-degree affected region [154]. PBS has numerous comorbidities (e.g., endometriosis), among which migraine is very frequent [11,155].

Keller et al. [156] explored the comorbid medical conditions of patients with bladder pain syndrome/interstitial cystitis (BPS/IC) in Taiwan using a cross-sectional study design and a population-based administrative database. The study included 9,269 subjects with BPS/IC and 46,345 randomly selected comparison subjects. Thirty-two medical comorbidities were considered (hypertension, congestive heart failure, cardiac arrhythmias, blood loss anemia, peripheral vascular disorders, stroke, ischemic heart disease, hyperlipidemia, hepatitis B or C, migraines, headaches, Parkinson’s disease, rheumatoid arthritis, systemic lupus erythematosus, ankylosing spondylitis, pulmonary circulation disorders, chronic pulmonary disease, diabetes, hypothyroidism, renal failure, fluid and electrolyte disorders, liver diseases, peptic ulcers, deficiency anemias, depressive disorder, psychoses, metastatic cancer, solid tumor without metastasis, alcohol abuse, drug abuse, and asthma) between subjects with and without BPS/IC.

With the exception of metastatic cancer, the BPS patients had a significantly higher prevalence of all the medical comorbidities analyzed, and thus also migraine and headache, than the patients without BPS.

The mechanisms beyond PBS still remain partially unknown, and the condition is not linked to a documentable organic cause but is probably contributed to by mechanisms of central sensitization [37,46], similarly to what happens in headache at a high frequency of crises/chronic. With this respect, it is worth noting that PBS also often co-occurs with other pain conditions characterized by the same phenomena of central sensitization, as testified by a diffuse decrease in pain thresholds, conditions such as irritable bowel syndrome or fibromyalgia. In this context, patients with more comorbidities present more severe and bothersome PBS symptoms [6].

The PBS–headache comorbidity has implications for therapy. Among the pharmacologic treatments for PBS are tricyclic antidepressants, particularly amitriptyline, but also anticonvulsants, both of which are also commonly employed for headache prophylaxis [157].

## 6. Calcitonin-Gene-Related Peptide (CGRP), Migraine, and Visceral Pain

CGRP is a neuropeptide of 37 amino acids produced from alternative RNA processing of the calcitonin gene. In humans, it has two major forms: a-CGRP, primarily involved in migraine (see below), which is mainly expressed in primary sensory neurons of the dorsal root ganglia, throughout the trigeminal system (located on Adelta-fibers and Cfibers) and in vagal ganglia; and b-CGRP, which is found mainly in intrinsic enteric gray neurons.

CGRP is a potent vasodilatator. It exerts both protective actions (at the cardiovascular level, in the processes of wound healing) and a pain signaling function (e.g., it enhances the release of substance P from primary afferent terminals, promoting nociceptive transmission, and it modulates the synaptic transmission of glutamate). Its key role in migraine pathophysiology is due to its vasodilatator capacity but also to its ability in modulating neuronal excitability, with the triggering and maintenance of peripheral and central sensitization, crucial phenomena in migraine [158].

The pathophysiology of migraine is, indeed, complex and involves several mechanisms [159,160]. Migraine-specific triggers in genetically predisposed people cause primary brain dysfunction with subsequent dilatation of cranial blood vessels. The dilatation mechanically activates sensory perivascular fibers of the trigeminal nerve, with nociceptive impulses then conveyed to the brainstem and higher brain centers and the release of vasoactive peptides, such as substance P (SP), a potent mediator of increased microvascular permeability and CGRP. Neurogenic inflammation then takes place with increased blood flow, edema, and the recruitment of inflammatory cells to the local area and with the degranulation of mast cells and the release of proinflammatory and inflammatory molecules. The process can activate meningeal nociceptors with a further increase in the level of activation of the sensory trigeminal fibers, perpetuating the release of vasoactive peptides, including CGRP. As migraine progresses, sensitization occurs in the spinal cord and brainstem.

Migraine pain then further increases, and hypersensitivity develops to environmental and other stimuli.

On this basis, it is not surprising that CGRP has generated increasing interest as a primary target for promising novel treatments for migraine pain, particularly monoclonal antibodies (mAbs) against CGRP. Anti-CGRP mAbs are macromolecules made of proteins that either directly target the ligand (CGRP), i.e., they bind to and neutralize the excessive CGRP released at perivascular trigeminal sensory nerve fibers, or target the CGRP receptor. The numerous studies so-far conducted with the available anti-CGRP monoclonal antibodies have shown satisfactory safety and efficacy outcomes in migraine prevention.

Several studies suggest that CGRP is also implicated in the pathophysiology of pain conditions other than headache. As reported by Schou et al. [161], there is an association between measured CGRP levels and somatic, visceral, neuropathic, and inflammatory pain, supporting the notion that CGRP may act as a neuromodulator in non-headache pain conditions [162,163,164,165]. Preclinical studies have established that CGRP plays a specific role in peripheral nociceptor sensitization [165,166]. Thus, as already underlined in previous sections, inhibition of CGRP through anti-CGRP mAbs, used for migraine prevention, would have the potential of being beneficial in visceral pain conditions where sensitization is documented, e.g., irritable bowel syndrome or PBS, only to mention some, therefore impacting two pain conditions (migraine and visceral pain) at the same time. Clearly, more studies are needed in this field, specifically examining sample patients affected with both migraine and visceral pain to be treated with anti-CGRP mAbs.

## 7. Visceral Pain and Headache: The Challenge of Comorbidity

The comorbidity of pain from internal organs and headache is not a simple co-occurrence, i.e., there is evidence that patients affected with the two conditions do not merely experience the sum of the symptoms of both. Instead, their symptomatology is often the result of a complex interaction of the two conditions. Visceral pain episodes often enhance the expression of headache, with headache attacks being triggered by visceral pain episodes; it has been shown, in fact, that the migraine or tension-type headache attacks tend to manifest within 24–48 h from a visceral painful episode from irritable bowel syndrome or suprapubic pain from PBS in comorbid patients [6]. This phenomenon is likely the result of central sensitization promoted by the visceral afferent input, which would favor the manifestation of the headache pain. In these patients, it has also been shown that effective treatment of the visceral pain condition (such as dietary treatment for IBS) also results in a reduction in the headache pain (number and intensity of headache attacks) in the long run [6]. In the comorbidity, however, the symptom enhancement is likely to be bi-directional. As reported in the previous sections, for instance, migraineurs with comorbid endometriosis are almost five times more likely to have a severe form of the visceral disease with more intense pelvic pain [134].

Here again, the common substrate for the association of visceral pain with headache, but also of other pain conditions such as fibromyalgia, seems to be central sensitization. Therapeutic interventions primarily targeted at reducing this state of neuronal excitability would probably be effective in reducing the whole pain burden of the comorbid patient. Future controlled studies specifically directed at evaluating the symptom profile of visceral pain–headache comorbid patients before and after appropriate therapies will hopefully shed more light onto mechanisms and implications of this interaction.

## 8. Conclusions

Headache is comorbid with several different forms of pain from internal organs belonging to the cardiovascular, gastrointestinal, and genitourinary systems [8,13,25,167]. The most frequent types of headache pain occurring in comorbidity with visceral pain are represented by migraine and tension-type headache, while no data are so-far available regarding the interaction of cluster headache with visceral pain. Migraine occurs more frequently in combination with pelvic pain, especially in women, and in cardiac pain, while tension-type headache more frequently occurs in GI pain, especially in the case of IBS. The mechanisms beyond this association are likely to be multiple. For the comorbidity between migraine and cardiovascular events, an important role is probably played by the promotion of endothelial dysfunction by the migraine condition, but oxidative stress and genetic and hormonal factors are also likely involved. In the other cases of comorbidity, sensitization phenomena are likely to represent a major contributor [146], i.e., peripheral sensitization at the visceral level could secondarily promote widespread central sensitization, which could enhance the triggering and transmission of headache pain. This would explain why headache attacks appear more frequently and severely in comorbidity patients when visceral pain is in place. Women are particularly subject to these phenomena, considering the relatively higher frequency than in men of visceral pain conditions associated with peripheral and central sensitization from the reproductive area (due to the higher complexity of the reproductive function in the female vs. male sex, which predisposes women to more numerous pains from this area) and the role played by estrogen in CNS sensitization.

The visceral pain–headache comorbidity appears particularly important from a clinical point of view, mostly for the notable implications for the evaluation and treatment of the complex patient. Shared mechanisms between the two conditions highlight, for instance, the possibility of therapeutic interventions able to control the symptoms from both diseases. We live in a highly specialized medical era where the tendency often prevails to focus separately on single-district clinical problems; a global and integrated medical assessment should instead always be performed in order to optimize treatment of the affected patients.

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
