# Peer review of "Pain from Internal Organs and Headache: The Challenge of Comorbidity"

_diagnostics, 2024, doi:10.3390/diagnostics14161750_

Round 1

Reviewer 1 Report

Comments and Suggestions for Authors

The manuscript "Pain from Internal Organs and Headache : The Challenge of 2 Comorbidity" is a search for the causes of pain.

My comments:

Introduction

2. section - The authors describe various causes of pain and connect various places and causes of pain. In my opinion, there is a lack of molecular basis, e.g. if pain arises in the heart for cardiological reasons, how does it reach the CNS, and then how do we feel it?

What molecular parameters are involved?

3. Conclusion - state what type of pain occurs most often, what it causes and what are its causes.

What organs cooperate and, for example, their effect will be migraine pain and when tension headaches, or cluster headaches?

What are the uses of antibodies against CGRP protein?

4. Some information should be summarized in tables and figures to make the manuscript more readable.

5. Are there any predictive and prognostic factors related to the development of pain and in what area?

6. The information contained in the manuscript needs to be supplemented and organized.

Author Response

REVIEWER 1.

Q: Introduction

2. section - The authors describe various causes of pain and connect various places and causes of pain. In my opinion, there is a lack of molecular basis, e.g. if pain arises in the heart for cardiological reasons, how does it reach the CNS, and then how do we feel it? What molecular parameters are involved?

A: A better explanation for various forms of visceral pain, cardiac pain in particular, is provided, with description of known molecular mechanisms, as requested (see sections on the specific visceral pain conditions)

Q: 3. Conclusion - state what type of pain occurs most often, what it causes and what are its causes.

What organs cooperate and, for example, their effect will be migraine pain and when tension headaches, or cluster headaches?

What are the uses of antibodies against CGRP protein?

A: The Conclusion has been implemented. The requested information has also been inserted in the preceding sections

Q: 4. Some information should be summarized in tables and figures to make the manuscript more readable.

A: We indeed have Tables 1 and 2 summarizing the main visceral pain – headache comorbidities and the main mechanisms of these comorbidities

Q: 5. Are there any predictive and prognostic factors related to the development of pain and in what area?

A: Not entirely sure what it is meant by this comment. In general, very frequent visceral attacks are predictive of an increased number of headache attacks since the repetition of the visceral algogenic input promotes the triggering of the headache episodes.

Q: 6. The information contained in the manuscript needs to be supplemented and organized.

A: We supplemented and re-organized the information in the manuscript according to the above request and also the requests by the other reviewers

Reviewer 2 Report

Comments and Suggestions for Authors

Thank you for the opportunity to review this paper.

My comments are below.

The first research question posed in this review concerns the possible relationship between headache and visceral pain: do they share any common pathophysiological aspect? This question is quite broad and not well defined. Although headaches and internal pain can occur together, they often do not. Therefore, answering these questions in a narrative review is problematic. It is not clear why the authors are trying to find some kind of connection between them, if a patient has internal pain, why can’t we just deal with it directly without worrying about the headache?

Another question posed in this article is: Is it a simple sum of individual pain conditions, or is there a reciprocal influence of symptoms between two pain conditions that has important implications for treatment? This question is also broad and not well defined as it is impossible to determine a true/universal answer.

Given that a lot of research has been done on this topic, as shown in the references, it is unclear what is the novelty of this narrative review?

Although the English is good, the review is difficult to read and I'm not sure it will be of interest to readers. This may be due to the style of work (narrative review). Perhaps the work should be rewritten in a different format.

Some other problems: the yellow marks in the text are distracting. Referencing like [5-26] on page 2, line 48 need clarification.

Novelty and scope: The research questions are too broad and not well defined.

Significance: Weak. What are the hypotheses?

Quality: The review quality could be improved. Think about how to improve your review. 

Scientific Soundness: I think this study needs to be redesinged and become more specific and targeted. 

Interest to the Readers: I don’t think the article will attract a wide readership.

Overall Merit: The overall benefit of publishing this work is limited.

English level: appropriate.

Author Response

REVIEWER 2

Q: The first research question posed in this review concerns the possible relationship between headache and visceral pain: do they share any common pathophysiological aspect? This question is quite broad and not well defined. Although headaches and internal pain can occur together, they often do not. Therefore, answering these questions in a narrative review is problematic. It is not clear why the authors are trying to find some kind of connection between them, if a patient has internal pain, why can’t we just deal with it directly without worrying about the headache?

A: Epidemiological studies clearly indicate that several forms of visceral pain are significantly more frequent in headache patients than in the general population. Based on these data the question consequently arises of possible common pathophysiological mechanisms between specific forms of visceral pain and headache. There is also evidence that this co-occurrence results in an enhancement of symptoms between the two conditions, and that treatment of one condition in comorbid patients has important repercussions on the other. We strongly believe that in comorbid patients, one cannot just deal with visceral pain directly without worrying about the headache, since the two conditions interfere with each other, and the relative treatments need to be coordinated and integrated. We realize that perhaps our explanation of these concepts was not clear in the manuscript and have therefore implemented these descriptions/explanations in the text.

Q: Another question posed in this article is: Is it a simple sum of individual pain conditions, or is there a reciprocal influence of symptoms between two pain conditions that has important implications for treatment?This question is also broad and not well defined as it is impossible to determine a true/universal answer.Given that a lot of research has been done on this topic, as shown in the references, it is unclear what is the novelty of this narrative review?

A: As stated above, there is evidence of interference of visceral pain and headache symptoms. We agree that the question is broad, but nevertheless very important for the repercussions it may have in the clinical settimg. It is true that many research studies have addressed the comorbidity issue, but mostly in regard to individual visceral pain conditions, and often to sectorial aspects of each. To the best of our knowledge this is the first comprehensive analysis of these concepts. We believe the novelty of this review is the attempt to find a synthesis of the various visceral pain-headache conditions interefering with each other, and focus on the importance of the comorbid patient as the center of an integrated diagnosis and treatment regimen.

Q: Although the English is good, the review is difficult to read and I'm not sure it will be of interest to readers. This may be due to the style of work (narrative review). Perhaps the work should be rewritten in a different format.

A: We acknowledge the comment by the reviewer, but, however, we feel we cannot adhere to this request in full. This would mean altering the nature of the review completely and also the messages we want to convey. We have, however, made significant implementation/modification in the various sections, which we believe will facilitate the reading

Q: Some other problems: the yellow marks in the text are distracting. Referencing like [5-26] on page 2, line 48 need clarification.

A: Not sure what the reviewer is referring to. There were no yellow marks in the text we submitted to the journal. In this revised version, instead, the text does contain parts which are highligheted in yellow to indicate the modified/implemented sections.

Reviewer 3 Report

Comments and Suggestions for Authors

The present review aimed to describe the association between headache and visceral pain their possible underlying mechanisms of the associations, and their diagnostic and therapeutic implications based on the literature. The review as a whole is good but needs minor revision to increase its quality.

1.      The author should describe the molecular association of headache with visceral pain concerning neuroanatomy.

2.      The author should also discuss the gender differences between overlapping pain conditions.

3.      Female reproductive organs and headache section author should discuss the role of sex hormones in comorbid pain.

4.   Authors should discuss the role of peripheral and central sensitization in the association of headaches with visceral pain.

Author Response

REVIEWER 3

The present review aimed to describe the association between headache and visceral pain their possible underlying mechanisms of the associations, and their diagnostic and therapeutic implications based on the literature. The review as a whole is good but needs minor revision to increase its quality.

Q:1. The author should describe the molecular association of headache with visceral pain concerning neuroanatomy.

A: Thank you for your important suggestion. We have implemented the text by addressing this aspect (see also our answers to the questions by Reviewer 1)

Q: 2.The author should also discuss the gender differences between overlapping pain conditions.

A: Overlapping pain conditions have now been discussed with respect to gender differences, as requested, specifically in the relative sections

Q: 3. Female reproductive organs and headache section author should discuss the role of sex hormones in comorbid pain.

A: The role of sex hormones has been addressed

Q: 4. Authors should discuss the role of peripheral and central sensitization in the association of headaches with visceral pain.

A: This role has now been discussed

Round 2

Reviewer 1 Report

Comments and Suggestions for Authors

The authors made numerous changes to the manuscript. I accept the revised manuscript.

Reviewer 2 Report

Comments and Suggestions for Authors

The aim of this paper is to describe recent headache and visceral pain associations, underlying mechanisms, diagnostic ways and therapeutic consequences based on the literature.

The major contribution: Headache was discussed from different aspects such as cardiovascular, gastrointestinal and genitourinary systems. It also takes into account the gender and age of the patients.

Value/Importance to the readers/science: In the manuscript it states that no data are so far available regarding interaction of cluster headache with visceral pain, so the area is new and there is a literature gap. So this manuscript gives some information  about it.

The problem with this manuscript is that it has a very high self-citation rate (Self-cited references:  2, 4, 6, 24, 27, 28, 29, 30, 31, 32, 33, 35, 36, 37, 38, 39, 40, 41, 42, 72, 150, 158, 159, 160   → 24 out of 167 (14.37%) references) and this needs explanation and careful review.

The authors even try to cite the unpublished result (line 912).

The manuscript seems to have issues with plagiarism (iThenticate showed 40% similarity rate), this should be double checked.

For example, the report shows in red paragraphs of text like this: "It is often associated with negative cognitive, behavioural, sexual and emotional consequences as well as with symptoms suggestive of lower urinary tract, sexual, bowel, pelvic floor or gynaecological dysfunction. In the case of documented nociceptive pain that becomes chronic/persistent through time, pain must have been continuous or recurrent for at least three months".

These issues should be carefully checked to avoid retraction of the paper in the future. 
